# The Arrangement of the Peripheral Olfactory System of *Pleuragramma antarcticum*: A Well-Exploited Small Sensor, an Aided Water Flow, and a Prominent Effort in Primary Signal Elaboration

**DOI:** 10.3390/ani12050663

**Published:** 2022-03-06

**Authors:** Stefano Aicardi, Matteo Bozzo, Andrea Amaroli, Lorenzo Gallus, Beatrice Risso, Erica Carlig, Davide Di Blasi, Marino Vacchi, Laura Ghigliotti, Sara Ferrando

**Affiliations:** 1Laboratory of Comparative Anatomy, Department of Earth, Environmental, and Life Sciences (DISTAV), University of Genoa, 16132 Genoa, Italy; stefano.aicardi94@libero.it (S.A.); galluslorenzo@gmail.com (L.G.); beatricerisso1997@gmail.com (B.R.); 2Laboratory of Developmental Neurobiology, Department of Earth, Environmental, and Life Sciences (DISTAV), University of Genoa, 16132 Genoa, Italy; matteo.bozzo@edu.unige.it; 3Department of Surgical and Diagnostic Sciences, University of Genoa, 16132 Genoa, Italy; andrea.amaroli.71@gmail.com; 4Department of Orthopedic Dentistry, Faculty of Dentistry, First Moscow State Medical University (Sechenov University), 119991 Moscow, Russia; 5Institute for the Study of the Anthropic Impacts and the Sustainability of the Marine Environment (IAS), National Research Council (CNR) of Italy, Via De Marini 6, 16149 Genoa, Italy; erica.carlig@ias.cnr.it (E.C.); davide.diblasi@szn.it (D.D.B.); marino.vacchi@ias.cnr.it (M.V.); 6Department of Integrative Marine Ecology (EMI), Stazione Zoologica Anton Dohrn, National Institute of Marine Biology, Ecology and Biotechnology, Genoa Marine Centre, 16126 Genoa, Italy

**Keywords:** Antarctic silverfish, olfactory rosette, olfactory nerve, olfactory bulb, fish olfaction, isotropic fractionator

## Abstract

**Simple Summary:**

How animals perceive their surrounding environment is crucial to their reactions and behavior. Olfaction, among others, is one of the more important senses for wide-range communication and in low-light environments. This study aims to give a morphological description of the peripheral olfactory system of the Antarctic silverfish, which is a key species in the coastal Antarctic ecosystem. The head of the Antarctic silverfish is specialized to assure that the olfactory organ keeps in contact with a large volume of water, even when the fish is not actively swimming. The sensory surface area and the number of neurons in the primary olfactory brain region show that this fish invests energy in the detection and elaboration of olfactory signals. In the cold waters of the Southern Ocean, the Antarctic silverfish is therefore likely to rely considerably on olfaction.

**Abstract:**

The olfactory system is constituted in a consistent way across vertebrates. Nasal structures allow water/air to enter an olfactory cavity, conveying the odorants to a sensory surface. There, the olfactory neurons form, with their axons, a sensory nerve projecting to the telencephalic zone—named the olfactory bulb. This organization comes with many different arrangements, whose meaning is still a matter of debate. A morphological description of the olfactory system of many teleost species is present in the literature; nevertheless, morphological investigations rarely provide a quantitative approach that would help to provide a deeper understanding of the structures where sensory and elaborating events happen. In this study, the peripheral olfactory system of the Antarctic silverfish, which is a keystone species in coastal Antarctica ecosystems, has also been described, employing some quantitative methods. The olfactory chamber of this species is connected to accessory nasal sacs, which probably aid water movements in the chamber; thus, the head of the Antarctic silverfish is specialized to assure that the olfactory organ keeps in contact with a large volume of water—even when the fish is not actively swimming. Each olfactory organ, shaped like an asymmetric rosette, has, in adult fish, a sensory surface area of about 25 mm^2^, while each olfactory bulb contains about 100,000 neurons. The sensory surface area and the number of neurons in the primary olfactory brain region show that this fish invests energy in the detection and elaboration of olfactory signals and allow comparisons among different species. The mouse, for example—which is considered a macrosmatic vertebrate—has a sensory surface area of the same order of magnitude as that of the Antarctic silverfish, but ten times more neurons in the olfactory bulb. Catsharks, on the other hand, have a sensory surface area that is two orders of magnitude higher than that of the Antarctic silverfish, while the number of neurons has the same order of magnitude. The Antarctic silverfish is therefore likely to rely considerably on olfaction.

## 1. Introduction

The olfactory system of jawed fishes has a general organization common to all the species, formed by paired olfactory chambers, but it comes in many different arrangements [1,2,3,4,5]. The peripheral olfactory organ ranges from a simple smooth surface to a complex multilamellar shape (often called a rosette), resulting in different olfactory organ surface areas (OSAs) available in a given volume. The sensory epithelium, where the olfactory receptor neurons (ORNs) are located, covers the olfactory organ surface—either completely or intermingling variously with the non-sensory epithelium [4]. Usually, each olfactory chamber—which contains the peripheral olfactory organ—is connected to the environment by two nostrils: an anterior one, which allows water to enter the chamber (incurrent nostril), and a posterior one, which allows water to exit (excurrent nostril). Otherwise, some teleosts are monotrematous: their olfactory chamber is connected to the environment by a single opening, that grants bidirectional water flow [2]. The olfactory chamber can be a simple cavity or present with branches, usually called accessory nasal sacs [1,4,6]. Nostril(s), the olfactory chamber, and, when present, accessory nasal sacs, together with the olfactory organ, build up a system of spaces where water flows; this system gives rise to zones with different pressures and current velocities [6]. Water transports odorants to the olfactory organ, which, as said, is covered by both sensory and non-sensory epithelia. The epithelium in turn is covered by mucus, in which the odorants dissolve before binding to the receptors on the cell membrane of ORNs [7]. The signals elicited by the binding between odorants and receptors are transmitted toward the central nervous system by the axons of the ORNs. Axons gather in the fila olfactoria, which form the olfactory nerve (ON), projecting to the part of the telencephalon that is the very first relay for olfactory information: the olfactory bulb (OB). The position of the OB is variable in teleosts—it can be sessile or pedunculated. In the case of a sessile OB, the ON has a certain length, while the olfactory tract, which connects the OB to the rest of the telencephalon, is short or virtually absent; in the case of pedunculated OB, the olfactory nerve is almost indistinguishable, as the fila olfactoria directly reach the OB, while the peduncle contains the olfactory tract. Additionally, a third intermediate condition has been described in some teleost species, referred to as pseudosessile [8]. In the OB, axons form a nervous layer, where they interweave so that each axon reaches its target glomerulus in the glomerular layer. Glomeruli are zones where the axons from the ON make contact with the dendrites from mitral cells. Mitral cells are large projection neurons, and their axons are the major component of the olfactory tract. Before forming the olfactory tract, the axons of mitral cells pass through a further layer, that of the granular cells. This organization in four layers of the OB is generally conserved in vertebrates [9,10]. A qualitative description of the above-mentioned structures is known for several teleost species [1,5,11,12,13,14]. Some authors attempted a more quantitative approach to the study of the anatomy of the peripheral olfactory system of teleosts, quantifying the epithelial surface area of the olfactory organ [15,16,17], the density of ORNs in the olfactory epithelium [5], the number of axons in the olfactory tract [18,19,20], the number of axons in the olfactory nerve [21,22,23,24], and the number of cells in the OB [16].

Notothenioidei is the only fish sub-order that has adapted to the coldest ocean on the Earth: the Antarctic Ocean. The morphology of at least some parts of the olfactory system of notothenioids has been investigated over the years [16,25,26,27,28,29,30,31,32,33]. The Antarctic silverfish *Pleuragramma antarcticum* is the dominant pelagic fish in coastal Antarctica and a keystone species in local ecosystems. Unlike most notothenioids, the Antarctic silverfish is associated with water columns, with eggs developing in a unique environment—platelet ice—and adults living in the midwaters over the continental shelf. Its pelagic lifestyle, unusual among the swim bladder-less Notothenioidei, is made possible by relevant morphological modifications, selected on an evolutionary timescale in a process of secondary pelagization. [34,35,36]. On the other hand, the importance of migrations around the Antarctic continent for this species seems clear. Although its life cycle has been only partially disclosed, a homing behavior was hypothesized for the Antarctic silverfish [37], with reproductive migrations driven by environmental cues from open waters to selected coastal areas. Even if it is not yet proven to date, it may be possible that homing is assisted by olfaction. The use of the specific name *P. antarcticum* follows the currently recognized change of nomenclature [38]. The OB of *P. antarcticum* has been the object of a study focused on the sensory areas of the brain [39]. In that article, the size of the OB was evaluated and compared to other brain regions and the OB of other species. 

This work aims to illustrate the anatomy of the peripheral olfactory system of adult *P. antarcticum*, describing the olfactory chamber, the olfactory organ, the ON, and the OB. 

## 2. Materials and Methods

### 2.1. Sampling and Tissues Preparation

The heads from six adult specimens of the Antarctic silverfish *Pleuragramma antarcticum* were obtained from two different operations. Two specimens were collected at Cape Hallett (Ross Sea, Antarctica) during the Western Ross Sea Voyage 2004 [40] onboard the RV Tangaroa. The other four were collected at Iselin Bank (Ross Sea, Antarctica) during the Tangaroa Ross Sea Voyage 2019 [41]. The size of the specimens was not recorded after capture and only the heads were fixed in paraformaldehyde 4%, rinsed in PBS, and stored in 70% ethanol for anatomical and histological investigations. To determine the size of the specimens a linear function between standard length (SL) and brain length (BL, from the olfactory bulb to the obex) was based on the dataset from [39]:SL = 20.811 BL − 97.568 (mm)(1)

To make comparisons with data from the literature where the total length (TL) of the fish is indicated, this linear function for the length was obtained for *P. antarcticum* from [42]:TL = 1.092 SL + 0.284 (cm)(2)

The calculated sizes of the *P. antarcticum* specimens are reported in Table 1 and Table 2.

At the time of the analysis, the heads were dissected to isolate the brain and the olfactory organ. Alternatively, the olfactory organ was not removed, and the histological analysis was performed on the anterior part of the snout after five days of decalcification in Osteodec (Bio-Optica, Milano, Italy). 

### 2.2. Gross Morphology

One olfactory organ for each specimen was analyzed through a stereomicroscope Leica DMRB (Leica Microsystems, Wetzlar, Germany) equipped with a Moticam 10+ camera (Motic Europe, Barcelona, Spain) to count the olfactory lamellae, to evaluate the OSA, and to measure the olfactory nerve thickness using ImageJ [43]. As specified, all the measurements were collected after fixation and storage in 70% ethanol.

### 2.3. Histology

The isolated olfactory organ and nerves, or the whole anterior part of the head, were dehydrated in ethanol, paraffin embedded, and 5 µm cut using a microtome, according to standard protocols. The slides were stained using Hematoxylin-Eosin, Masson’s Trichrome, and Azan Trichrome (Bio-Optica, Milano, Italy). The stained sections were observed using a transmitted light microscope Leica DMRB equipped with a Moticam 3+ (Motic Europe, Barcelona, Spain) or through a transmitted light microscopy Olympus BX60 equipped with a Microvisioneer (Esslingen am Neckar, Germany) camera and acquisition software.

### 2.4. Isotropic Fractionator

Each OB (one hemisphere) was weighed and analyzed according to the isotropic fractionator technique for cell and neuron counting [44]. Briefly, each OB was homogenized in 1% Triton in 40 mM trisodium citrate. After centrifugation at 10,000 rpm for 45 minutes, the pellet, which contained all the extracted nuclei, was resuspended in a known volume of phosphate buffer saline (pH 7.4) and 4′,6-diamidino-2-phenylindole (DAPI) for nuclei detection. The solution with stained nuclei was continuously stirred to become isotropic, and samples from the solution were used for nuclei counting in a hemocytometer under an epifluorescence microscope (Leica DMRB) equipped with a Moticam 3+ (Motic Europe, Barcelona, Spain). The total number of nuclei in the solution, and thus in the OB, was calculated. An aliquot of extracted nuclei solution was treated for immunocytochemistry using a mouse anti-Neuronal Nuclei (anti-NeuN) antibody (Millipore, Darmstadt, Germany, Mab377, 1:100) overnight at room temperature. NeuN is a neuronal marker and can be used as a marker for neuronal nuclei in vertebrates [45,46,47]. After rinsing in PBS and centrifuging at 10,000 rpm for 3 min, the pellet was incubated with a secondary antibody goat anti-mouse conjugated with the fluorochrome Dylight^®^ 488 (Immunoreagent, Raleigh, NC, USA, GtxMu-003-D488NHS). At least 100 nuclei were evaluated both in UV light for DAPI and in blue light for the Dylight^®^ 488, and the percentage of neurons and non-neuronal cells was evaluated. The mitral cells, large interneurons present in the OB of vertebrates, are one of the few neuronal populations with a NeuN-negative cell nucleus [48]. Nevertheless, the number of mitral cells has been considered negligible for the evaluation of neuron number in the OBs—at least in mammals [49].

### 2.5. Building of the Dataset 

The quantitative measurements and counts obtained for the adults were then compared with data from other teleosts acquired from the literature. Teichmann [17] reports the OSA for 11 species of teleost fish. One of those species is the European eel, *Anguilla anguilla*, the OSA of which was also evaluated in another article [15]. As the average size of the fish considered (Teichmann [17]: average SL 51 cm; Atta [15]: average SL 54 cm) and the average OSA evaluated (Teichmann [17]: average OSA for two olfactory organs 575 mm^2^; Atta [15]: average OSA for two olfactory organs 424 mm^2^) are not identical in the two articles, but similar enough, we here decided to consider numbers from [17], for homogeneity with the other 10 species. In [17], several individuals for each species were analyzed and the average values for standard length and whole OSA are also given. Ferrando et al. [16] report the total length and the OSA of one olfactory organ of one specimen of the nototheniid species: *Dissostichus mawsoni*. Data from the present paper, from [16], and from [17] are gathered in Table 1. It is noteworthy that all the measures are average values from more than one specimen of the same species, except for *D. mawsoni*, where only one specimen was considered. Moreover, the body size reported for the specimen of *D. mawsoni* is the total length while, for all other fishes, the standard length is indicated.

The teleost species used in Table 1 for the comparison with *P. antarcticum* were chosen because of the availability of data from the literature. Many are mainly freshwater species (*Phoxinus phoxinus*, *Gobio gobio*, *Squalius cephalus*, *Tinca tinca*, *Nemachilus barbatulus*, *Exos Lucius*, *Lota lota*, *Perca fluviatilis*), two species are fully or partially anadromous (respectively *Salmo irideus* and *Gasterosteus aculeatus*), one species (*Anguilla anguilla*) is catadromous, and one species is marine (*D. mawsoni*). Beside the anadromous and catadromous species, some others are characterized as *P. antarcticum*, by their spawning migrations during their lifetime (*S. cephalus, P. fluviatilis, D. mawsoni).* For only some species is occurrence in groups documented (*S. cephalus*, *G. aculeatus*) [50].

In the literature, the isotropic fractionator technique has been applied to the OB of only one specimen of a teleost species: *D. mawsoni*. Unfortunately, the anti-NeuN antibody failed to work on that species, so the number of cells in the OB of that specimen is known, but not the percentage (and number) of neurons [16]. Considering non-teleost fish, the isotropic fractionator technique has been applied, to date, to the OB of two catshark species: *Scyliorhinus canicula* and *Galeus melastomus* [51]. Here, the number of cells and the cell density in the OB of *P. antarcticum* are compared to the same parameters in *D. mawsoni* and the two catshark species. Moreover, the number of neurons, the neuron density, and the ratio of other cells/neurons in the OB of *P. antarcticum* are compared to the same parameters in the two catsharks. In [51], another interesting parameter was also calculated: the number of neurons in the OB normalized to square millimeters of OSA. This parameter, which puts together two variables from the sensor and the first relay, promises to be quite interesting when available for several vertebrate species. We here calculated it for adult *P. antarcticum*. The measured and calculated numbers are presented in Table 2. Some scatterplots were obtained using the ggplot2 R package [52], to present graphically the data in Table 1 and Table 2.

**Table 2 animals-12-00663-t002:** Measurements and counts for 4 species of fish from the present study and the literature. TL = the total length was measured for fresh fish for all the specimens but *P. antarcticum* specimens, where it was calculated on the basis of paraformaldehyde-fixed and ethanol 70%-preserved brains (See Materials and Methods). RoL = Rosette Length; RoW = Rosette Width; LN = lamellar number for one rosette; ES = epithelial surface area for one rosette; OB = mass of the olfactory bulb (one hemisphere); OB cells = number of cells in one OB; OB neu = number of neurons in one OB; Neu% = percentage of neurons in the OB cells; Cell/mg = number of cells per milligram of OB tissue; Neu/mg = number of neurons per milligram of OB tissue.

Species	Specimen	TL (mm)	Sex	RoL (mm)	RoW (mm)	LN	ES (mm^2^)	OB (mg)	OB Cells	OB neu.	Neu%	Cell/mg	Neu/mg	Cell/ mm^2^	Neu/ mm^2^	Source
*P. antarctica*	17.2	150	-	-	-	-	15	1	1.89 × 10^5^	1.56 × 10^5^	82.5	1.89 × 10^5^	1.56 × 10^5^	1.23 × 10^4^	1.01 × 10^4^	present study
*P. antarctica*	17.3	150	-	2.6	1.8	23	22	1	2.80 × 10^5^	2.10 × 10^5^	75.1	2.80 × 10^5^	2.10 × 10^5^	1.29 ×10^4^	9.7 × 10^3^	present study
*P. antarctica*	20.1	211	F					1	1.41 × 10^5^	9.50 × 10^4^	67.2	1.41 × 10^5^	9.50 ×10^4^			present study
*P. antarctica*	20.2	184	F	3.2	2	24	24									present study
*P. antarctica*	20.3	181	M	3.2	2	27	26	1	1.71 × 10^5^	1.53 × 10^5^	89.1	1.71 × 10^5^	1.53 × 10^5^	6.59 × 10^3^	5.87 × 10^3^	present study
*P. antarctica*	20.4	214	F	3.3	2.3	24	32	1	7.70 × 10^4^	4.87 × 10^4^	63.2	7.70 × 10^4^	4.87 × 10^4^	2.43 ×10^3^	1.53 × 10^3^	present study
*D. mawsoni*	Dm4	1290	F	11	12	42	1041									[16]
*D. mawsoni*	Dm5	1210	M	12	13	41		30	3.60 × 10^6^			1.20 × 10^5^				[16]
*G.melastomus*	G1	360	F				3731	70	2.14 × 10^6^	1.64 × 10^6^	76.5	3.06 × 10^4^	2.34 × 10^4^	5.74 × 10^2^	4.40 × 10^2^	[49]
*G.melastomus*	G2	300	M				3113	55	2.00 × 10^6^	1.48 × 10^6^	72.9	3.64 × 10^4^	2.70 × 10^4^	6.4 × 10^2^	4.75 × 10^2^	[49]
*G.melastomus*	G3	270	M				2804	31	1.29 × 10^6^	8.30 × 10^5^	64.1	4.16 × 10^4^	2.66 × 10^4^	4.60 × 10^2^	2.96 × 10^2^	[49]
*G.melastomus*	G4	180	F				1877	20	8.50 × 10^5^	7.00 × 10^5^	82.7	4.25 × 10^4^	3.51 × 10^4^	4.53 × 10^2^	3.73 × 10^2^	[49]
*G.melastomus*	G5	170	F				1774	12	5.60 × 10^5^	4.40 × 10^5^	78.8	4.67 × 10^4^	3.68 × 10^4^	3.16 × 10^2^	2.48 × 10^2^	[49]
*G.melastomus*	G6	150	J				1568	9	1.90 × 10^5^	1.50 × 10^5^	80	2.11 × 10^4^	1.66 × 10^4^	1.21 × 10^2^	9.57 × 10^1^	[49]
*G.melastomus*	G7	135	J				1413	5	1.30 × 10^5^	1.00 × 10^5^	77.1	2.60 × 10^4^	2.02 × 10^4^	9.20 × 10^1^	7.08 × 10^1^	[49]
*G.melastomus*	G8	135	J				1413	6	1.70 × 10^5^	1.30 × 10^5^	75	2.83 × 10^4^	2.15 × 10^4^	1.20 × 10^2^	9.20 × 10^1^	[49]
*G.melastomus*	G9	150	M				1568	13	5.40 × 10^5^	3.70 × 10^5^	69.8	4.15 × 10^4^	2.87 × 10^4^	3.44 × 10^2^	2.36 × 10^2^	[49]
*G.melastomus*	G10	130	J				1362	7	2.80 × 10^5^	2.10 × 10^5^	76.9	4.00 × 10^4^	3.05 × 10^4^	2.06 × 10^2^	1.54 × 10^2^	[49]
*G.melastomus*	G11	110	J				1156	3	1.10 × 10^5^	7.00 × 10^4^	62.8	3.67 × 10^4^	2.22 × 10^4^	9.52 × 10^1^	6.06 × 10^1^	[49]
*S. canicula*	S1	480	M				4081	99	2.45 × 10^6^	1.85 × 10^6^	75.7	2.47 × 10^4^	1.87 × 10^4^	6.00 × 10^2^	4.53 × 10^2^	[49]
*S. canicula*	S2	330	M				2370	48	1.23 × 10^6^	9.40 × 10^5^	76.6	2.56 × 10^4^	1.96 × 10^4^	5.19 × 10^2^	3.97 × 10^2^	[49]
*S. canicula*	S3	210	M				1001	19	4.50 × 10^5^	3.30 × 10^5^	75	1.74 × 10^4^	1.76 × 10^4^	4.50 × 10^2^	3.30 × 10^2^	[49]
*S. canicula*	S4	200	F				888	21	3.50 × 10^5^	2.70 × 10^5^	76.5	1.76 × 10^4^	1.27 × 10^4^	3.94 × 10^2^	3.04 × 10^2^	[49]
*S. canicula*	S5	285	M				1857	36	1.22 × 10^6^	9.30 × 10^5^	76.2	3.36 × 10^4^	2.59 × 10^4^	6.57 × 10^2^	5.01 × 10^2^	[49]
*S. canicula*	S6	215	F				1958	23	3.30 × 10^5^	2.50 × 10^5^	74.6	3.96 × 10^4^	1.07 × 10^4^	1.69 × 10^2^	1.28 × 10^2^	[49]
*S. canicula*	S7	210	F				1001	22	3.70 × 10^5^	2.80 × 10^5^	74.1	2.86 × 10^4^	1.25 × 10^4^	3.70 × 10^2^	2.80 × 10^2^	[49]
*S. canicula*	S8	355	M				2655	49	1.21 × 10^6^	9.30 × 10^5^	76.5	2.31 × 10^4^	1.89 × 10^4^	4.56 × 10^2^	3.50 × 10^2^	[49]
*S. canicula*	S9	305	M				2085	42	9.10 × 10^5^	6.80 × 10^5^	74.1	3.64 × 10^4^	1.61 × 10^4^	4.36 × 10^2^	3.26 × 10^2^	[49]
*S. canicula*	S10	275	F				1742	23	6.30 × 10^5^	4.80 × 10^5^	76	5.35 × 10^4^	2.09 × 10^4^	3.62 × 10^2^	2.76 × 10^2^	[49]
*S. canicula*	S11	275	F				1742	28	1.13 × 10^6^	7.50 × 10^5^	66	3.43 × 10^4^	2.67 × 10^4^	6.49 × 10^2^	4.31 × 10^2^	[49]
*S. canicula*	S12	290	F				1914	26	1.53 × 10^6^	1.05 × 10^6^	68.3	2.42 × 10^4^	4.02 × 10^4^	7.99 × 10^2^	5.49 × 10^2^	[49]
*S. canicula*	S13	280	M				1799	33	1.23 × 10^6^	9.20 × 10^5^	74.7	3.73 × 10^4^	2.78 × 10^4^	6.84 × 10^2^	5.11 × 10^2^	[49]
*S. canicula*	S14	270	M				1685	24	9.60 × 10^5^	7.40 × 10^5^	76.7	4.00 × 10^4^	3.08 × 10^4^	5.70 × 10^2^	4.39 × 10^2^	[49]
*S. canicula*	S15	245	M				1400	22	6.30 × 10^5^	4.40 × 10^5^	69	2.86 × 10^4^	1.98 × 10^4^	4.50 × 10^2^	3.14 × 10^2^	[49]
*S. canicula*	S16	250	F				1457	17	7.00 × 10^5^	5.30 × 10^5^	76.2	4.12 × 10^4^	3.12 × 10^4^	4.80 × 10^2^	3.64 × 10^2^	[49]
*S. canicula*	S17	225	M				1172	20	7.30 × 10^5^	5.30 × 10^5^	72.4	3.65 × 10^4^	2.63 × 10^4^	6.23 × 10^2^	4.52 × 10^2^	[49]

## 3. Results

### 3.1. The Olfactory Organ of Adult P. antarcticum Is an Asymmetrical Rosette

The olfactory organ of the adult specimens of *P. antarcticum* (average TL 18.2 ± 2.6 cm) was an olfactory rosette characterized on average by 24 ± 2 lamellae, which were unevenly arranged in two rows along a central raphe. The length of the raphe was about 3.1 ± 0.3 mm in the specimens analyzed here, and it ran cranio-caudally. As the rosette was obliquely oriented within the olfactory chamber, one the parasagittal plane, the medial row of lamellae was dorsomedial, while the lateral one could be said to be ventrolateral. The ventrolateral row had two more lamellae than the dorsomedial, giving the olfactory rosette an asymmetrical shape that is not commonly seen in fish (Figure 1a and Figure 2a,b). Only one of the observed rosettes had a difference of three lamellae instead of two between the ventrolateral and the dorsomedial row. 

### 3.2. The Nasal Chamber Has a Reinforced Roof and Is Connected to Accessory Nasal Sacs

The cranial part of the olfactory rosette, corresponding to the dorsomedial row of the lamellae, was connected to the roof of the olfactory chamber, anterior to the opening of the nostril (Figure 2a,c). Posteriorly to this point of connection, the rosette was anchored only basally, and it stood free in a quite large olfactory chamber, opened dorsally in a single nostril (Figure 2a,d). Posteriorly to the nostril opening, the roof of the olfactory chamber—which overhung the rosette—was characterized by particularly compact connective tissue (Figure 3a,b and Appendix A). The main olfactory chamber was in continuity with three accessory nasal sacs. One accessory nasal sac was dorso-medially developed, and was quite small with respect to the other two; it could also have had the role of a branch of the main olfactory chamber (Figure 3a). The second sac was ventro-medially developed, presented with some diverticula, and reached the roof of the oral cavity—being separated from it only by a thin layer of tissue (Figure 3a,c). The third accessory nasal sac was ventro-lateral (Figure 3a).

### 3.3. The Olfactory Organ Surface Area of an Adult P. antarcticum Is Almost 50 mm^2^

The sensory olfactory epithelium covered almost the whole surface of the olfactory organ alongside the flat faces of the lamellae and the interlamellar curves. It had a thickness of 38.4 ± 0.6 µm. The non-sensory epithelium was localized along the free edge of the lamellae (Figure 4). Measuring the surface area of the flat faces of each lamella of one olfactory organ in order to obtain the OSA also provided a good evaluation of the sensory surface area (SSA). The measured OSA in one olfactory rosette from the adult specimens analyzed here ranged from 15 to 32 mm^2^. Each fish had, overall—on average—47.4 ± 11.9 mm^2^ of OSA, and probably of SSA too. All the measures are reported in Table 2. In Figure 5, the average value for *P. antarcticum* is plotted with measurements from the literature on other teleost species (see Materials and Methods).

### 3.4. In P. antarcticum the Olfactory Nerve Is Long, while the Olfactory Tract Is Short

The lamellae had a thin lamina propria where the bundles of the axons of the olfactory sensory neurons, i.e., the fila olfactoria, were visible. The bundles from each lamella gathered in a ribbon of fibers at the base of each lamellar row. More deeply, under the raphe, the two ribbons of nerve fibers joined and formed the olfactory nerve that ran toward the CNS (Figure 6). The olfactory nerve had a diameter of about 400 µm and it appeared to be divided into bundles (Figure 6c). Each olfactory nerve reached one of the olfactory bulbs. Olfactory bulbs can be considered sessile in *P. antarcticum* as they had an extremely short olfactory tract connecting them to the rest of the telencephalon.

### 3.5. In the Olfactory Bulb of P. antarcticum, Many Cells Elaborate the Signal from a Small Olfactory Organ

In the specimens analyzed, the average mass of one OB (one hemisphere) was 1 ± 0.5 mg. The numbers of cells and neurons in each OB of *P. antarcticum* are reported in Table 2, together with data from [16] regarding *D. mawsoni*, and from [51] regarding two catsharks. The mass of the OB and the number of cells and neurons (Figure 7a) had the same order of magnitude in *P. antarcticum* and catsharks of comparable body size. The number of neurons in the OB normalized for the OSA, instead, ranges in catsharks from 102 to 103, while it reached 104 in *P. antarcticum*.

## 4. Discussion

The peripheral olfactory organ of *P. antarcticum* is a rosette, which is a quite common shape in fish in general. The shape of the raphe, which is elongated, and the distribution of the lamellae, in two rows connected in the posterior part, correspond to the G arrangement, as indicated by [5]. Nevertheless, the olfactory rosette of *P. antarcticum* could be considered a modified G-type, because of the asymmetry of the two rows of lamellae. 

The lamellar number reported here (which was, on average, 24 for each rosette and 48 for the whole adult specimen) is likely to increase ontogenetically, and thus we can expect to find a lower number in younger specimens. The ontogenetic increase is expected because lamellae have a smooth surface, being without secondary folds on their surface, and a positive correlation between the number of olfactory lamellae and body length is found in fish without secondary folds [13]. Eastman [25] reported a lamellar number of 22–26 for this species, but without specifying the range of body sizes of the analyzed specimens. 

The presence of accessory nasal sacs, which are also close to the roof of the oral cavity, suggests a possible pumping mechanism of the sacs to create the water current in the olfactory chambers, exploiting mouth and head movements. In fact, the accessory nasal sacs are often described in monotrematous species, such as *P. antarcticum*, and are present in all notothenioids [32,33].

The OSA, and the SSA, were on average 24 mm^2^ for each rosette, and 48 mm^2^ for the organism. To evaluate if this is a large or small area, a comparison with other vertebrates was needed. A comparison with other teleost species from the literature [16,17] is shown in Table 1 and Figure 5. Data from [17] and from [16] regard the OSA, because the distribution of the sensory and non-sensory epithelium was examined in those articles, and the surface areas regard the lamellae. Considering the OSA, *P. antarcticum* was out of the confidence interval calculated for the regression curve, and its surface area was lower than expected for a teleost of that body size (Figure 5). The sensory and non-sensory epithelium coverage on the olfactory lamellae of teleosts was described by [5] and divided into four types. Type I is similar to that seen in *P. antarcticum* and shows the sensory epithelium continuously covering the sides of the lamellae, leaving the non-sensory epithelium on the edge of the lamellae. Types II, III, and IV instead show a more fragmentary distribution of the sensory epithelium. Yamamoto [5] described these different types in several teleost species, none of which are present in our dataset. The four species of Anguilliformes analyzed by [5] showed a distribution of type I; the same was seen for the two species of Gadiformes and the five species of Cypriniformes. This could suggest that, for at least some of the species in Figure 5, the OSA is a good proxy for the SSA—which is more informative about sensory function. In Figure 5, the species are indicated with different colors according to their order. As the number of species is relatively low, and some orders are represented by only one species, it would be difficult to draw possible correlations between ecological tracts—such as the migratory behavior that characterize some of the species—and the OSA.

The SSA was evaluated for other vertebrates in the literature. In Chondrichthyes, species with a body size ranging from 210 to 2300 mm had an SSA ranging from about 2000 to 120.000 mm^2^ [53]. The comparison with non-fish vertebrates could be performed only on a body-mass base. The body mass of the *P. antarcticum* specimens analyzed here was not measured, but they could be grossly calculated using the regression function from [42]:W(g) = 0.00170 × SL3.36 (cm)(3)

Thus, the *P. antarcticum* specimens analyzed here had a calculated body mass of 20–32 g. In mammals, the SSA has been measured in many species [54,55,56,57,58,59]. Although a review of the SSA of mammals is out of the scope of this work, from the partial literature cited, several mammalian species have a total SSA of the same order of magnitude as that of *P. antarcticum*: 2 species from the order Afrosorcida (range of SSA about 51–82 mm^2^, range of average body mass about 80–100 g), 2 species from Chiroptera (range of SSA about 32–68 mm^2^, range of average body mass about 16–28 g), 12 species from Eulipotyphla (range of SSA about 52–92 mm^2^, range of average body mass about 6–35 g), 1 species from Primates (SSA about 55 mm^2^, average body mass about 246 g), and 29 species from Rodentia (range of SSA about 29–104 mm^2^, average body mass about 7–195 g). All these numbers, regarding the SSA and the body mass, were obtained from the literature [54,57,58,60,61]. The SSA of *P. antarcticum*, which is two orders of magnitude lower than that of Chondrichtyes of the same body size, is one order of magnitude lower than expected among teleosts, but it is quite similar to mammalian species with a comparable body mass.

The comparison of SSAs brought up other parameters that should be considered: the ORN density (i.e., ORNs per mm^2^)—which can be reflected by the olfactory epithelium thickness and also by the cell size—the number of sensory cilia or microvilli that the ORNs bear, and the level of expression of olfactory receptors—which represent the actual part of the “sensory surface” that the odorant molecules link to. 

The thickness of the ON should reflect the number of ORNs and could allow other comparisons with other species; this requires some assumptions that are not certain—for example, that all the axons have the same diameter, and that the proportion of connective tissue among the bundles is similar. The ON diameter in a *D. mawsoni* specimen of 143 cm TL was evaluated to be 2–2.5 mm, and so 5–6 times larger than that of the adult *P. antarcticum* presented here. A quantitative analysis of the ON would require a transmitted electron microscope (TEM) observation as, for example, was carried out for *Esox lucius* [23]. 

At least for mammals, the absolute and relative size of the OB as a proxy of the olfactory capability of a species has been questioned [62]. The number of neurons has been proposed as a more reliable parameter to consider [48,60]. In adult *P. antarcticum*, the number of neurons in both the OBs was 2.65 × 10^5^ ± 1.24 × 10^5^. The mouse *Mus musculus*, whose SSA of about 47 mm^2^ [57] is roughly the same as *P. antarcticum*, has 3.89 × 10^6^ ± 1.25 × 10^6^ neurons in the two OBs [48]. In *M. musculus*, both OB mass (about 2 mg in *P. antarcticum* and about 14 mg in *M. musculus*) and the number of neurons in the OB, are one order of magnitude higher than in *P. antarcticum*. It is noteworthy that *M. musculus* is considered a macrosmatic species—at least among mammals. The comparison of *P. antarcticum* to the two catsharks [51], which are the only fish species for which the number of neurons in the OB is available, shows that this number is comparable in fish with a similar SL and comparable OB mass (Figure 7 and Table 2), although those fishes have a notably larger SSA (Table 2). This difference in SSA is shown also in Figure 7, where the number of neurons in the OB of *P. antarcticum*, normalized to the mm^2^ of surface area of the organ, was remarkably higher than those of the catsharks. This suggests that a large number of neurons elaborate the information coming from a relatively small (compared to catsharks) sensor in *P. antarcticum*.

## 5. Conclusions

Overall, the quantitative anatomy of the olfactory system of *P. antarcticum* presented in this study could indicate that this species relies on olfaction for important life tasks. Particularly, the presence of a number of neurons in the OB of this species, equal to that of catsharks and only tenfold less than that of macrosmatic mammals, suggests the necessity of high efficiency in odor first elaboration. Considering that the “expensive tissue hypothesis” [63] has been verified in teleosts too [64], the advantage of an efficient odor elaboration is likely to counterbalance the high energy cost of neural cells. 

## Figures and Tables

**Figure 1 animals-12-00663-f001:**
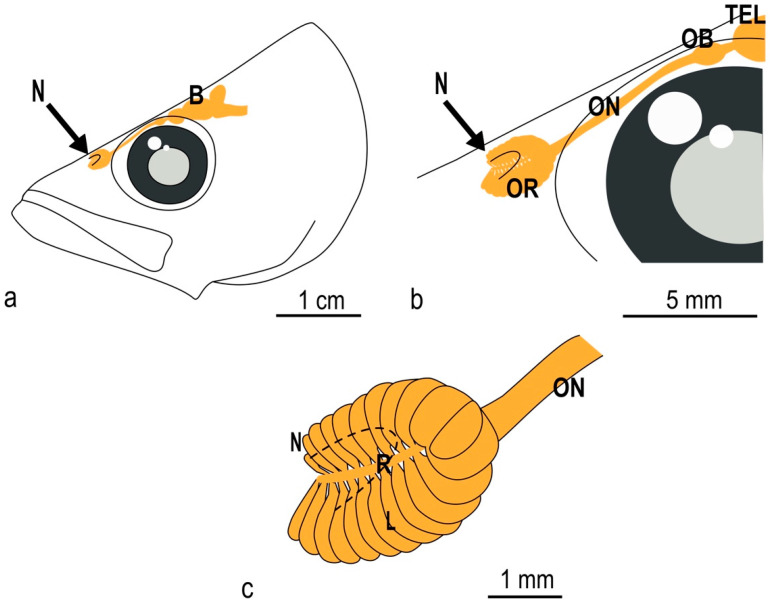
Schematic drawing of the adult *P. antarcticum* head with olfactory organ and brain. (**a**) Each nostril has a single, dorsally located opening. The position of the olfactory organ, nerve, bulb, and of the rest of the brain (in orange) is shown as if the head were transparent. (**b**) The nostril is open at the level of the anterior part of the rosette. The olfactory nerve runs along the orbit before entering the olfactory bulb. (**c**) The olfactory rosette has two unequal rows of lamellae; the shorter one is located medio-dorsally. B = brain; L = lamella; N = nostril; OB = olfactory bulb; ON = olfactory nerve; OR = olfactory rosette; R = raphe.

**Figure 2 animals-12-00663-f002:**
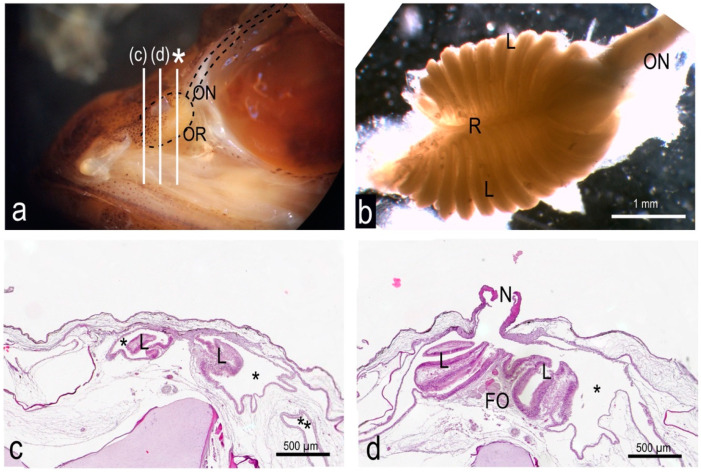
Gross morphology and histology of the olfactory organ of *P. antarcticum*. (**a**) Partially dissected head with the olfactory rosette visible in place in the olfactory chamber. The three white lines indicate the cutting plane in (**c**,**d**), and in Figure 3 (asterisk). (**b**) A dissected olfactory rosette showing the two unequal rows of lamellae. Anteriorly, the slightly lacerated tissue where the rosette was attached dorsally to the roof of the nasal chamber is visible. (**c**) Histological section of the olfactory chamber (asterisks) with the olfactory rosette, anterior to the opening of the nostril. The olfactory rosette is fused ventrally to the floor of the chamber and dorsally to its roof. The lamellae are not enveloped in a capsule nor fused with the wall of the olfactory chamber. A branch of the accessory nasal sac is visible in the low-right corner of the picture (double asterisk). (**d**) Histological section of the olfactory chamber with olfactory rosette, at the level of the opening of the nostril. The olfactory rosette is attached only to the floor of the olfactory chamber. FO = fila olfactoria; L = lamellae; N = nostril; ON = olfactory nerve; OR = olfactory rosette; R = raphe.

**Figure 3 animals-12-00663-f003:**
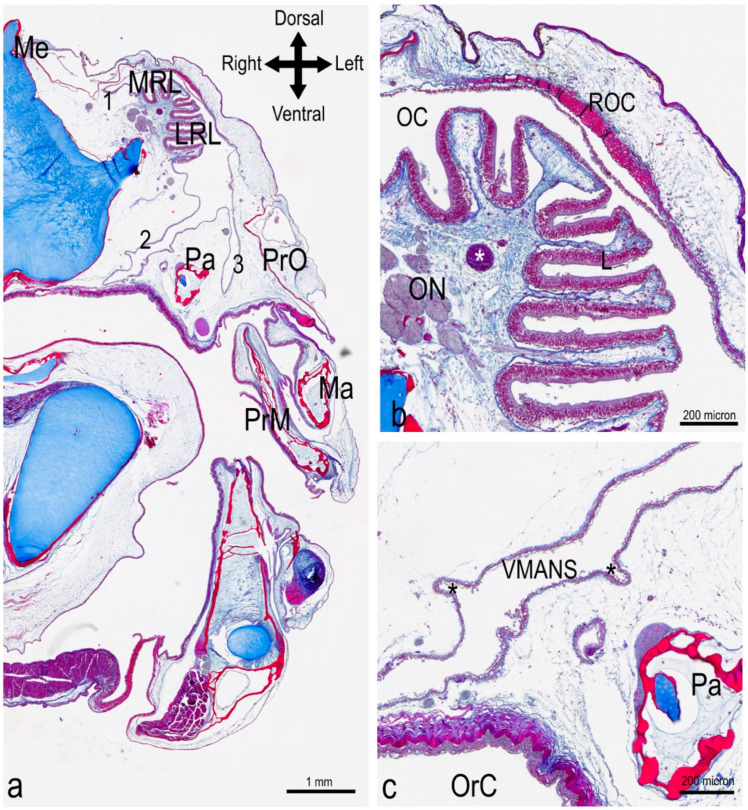
Histology of the head of *P. antarcticum*. The complete scan of this histological section at 100× is available in Appendix A. Transverse cutting plane. Azan Trichrome. (**a**) Complete section of half head; level indicated by the third line (with asterisk) in Figure 2a. The olfactory rosette is obliquely positioned in the nasal chamber, with the medial row of lamellae (the shorter row) being dorsal to the lateral (and longer) row of lamellae. The olfactory chamber is in continuity with accessory nasal sacs, which are narrow and branched. The ventromedial accessory nasal sac (number 2) runs internally to the palatine bone and reaches the roof of the mouth, being divided from the oral cavity by the oral mucosa. The ventrolateral accessory nasal sac (number 3) runs between the palatine and the preorbital bones. (**b**) Detail of the photograph in (**a**). The roof of the main nasal chamber is characterized by a thicker layer of fibrous connective tissue, compared to the rest of the wall of the main and accessory nasal sac. A large blood vessel (asterisk) is visible running along the raphe. (**c**) Detail of the photograph in (**a**). The ventromedial accessory nasal sac presents some diverticula of unknown function (asterisks). LRL = lateral row of lamellae; Ma = Maxilla; Me = mesethmoid; MRL = medial row of lamellae; OC = olfactory chamber; OrC = oral cavity; ON = olfactory nerve; Pa = Palatine bone; PrM = premaxilla; PrO = Preorbital bone; ROC = roof of the olfactory chamber; VMANS = ventromedial accessory nasal sac.

**Figure 4 animals-12-00663-f004:**
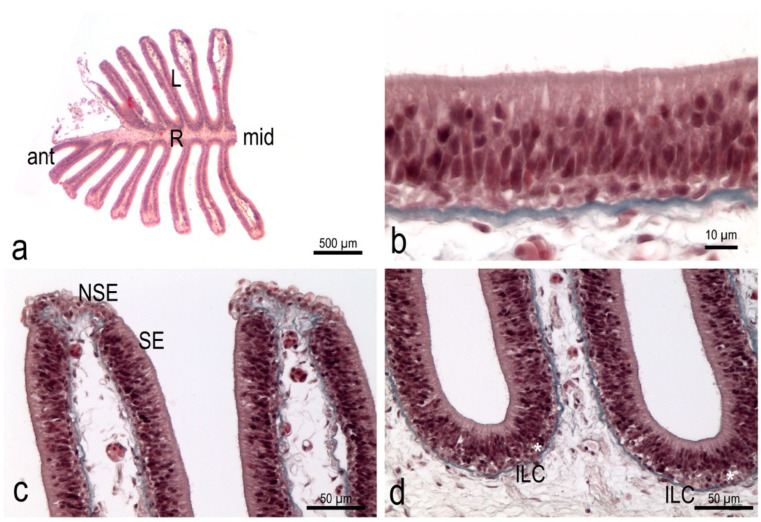
Histology of the olfactory rosette of *P. antarcticum*. (**a**) Hematoxylin-Eosin of the anterior half of an olfactory rosette. The cutting plane is parallel to the dorsal surface of the rosette, as seen in Figure 2b. (**b**) Masson’s Trichrome; detail of the sensory epithelium along the lamellae surface. (**c**) Masson’s Trichrome; apical part of two lamellae. The sensory epithelium covers the lamellae and only the free edge of each lamella shows a different, non-sensory epithelium. The connective tissue of the lamina propria is quite loose, and the blue of the collagen—indicating a more compact tissue—runs in a thin layer just under the basal lamina. (**d**) Masson’s Trichrome; basal part of two lamellae and one interlamellar curve. The basal layer of the sensory epithelium, in the interlamellar curve, is thicker than along the lamellae and shows at least two layers of round nuclei (asterisk). ant = anterior part of the rosette; ILC = interlamellar curve; L = lamella; mid = middle part of the rosette in an anterior-posterior direction; NSE = non-sensory epithelium; R = raphe; SE = sensory epithelium.

**Figure 5 animals-12-00663-f005:**
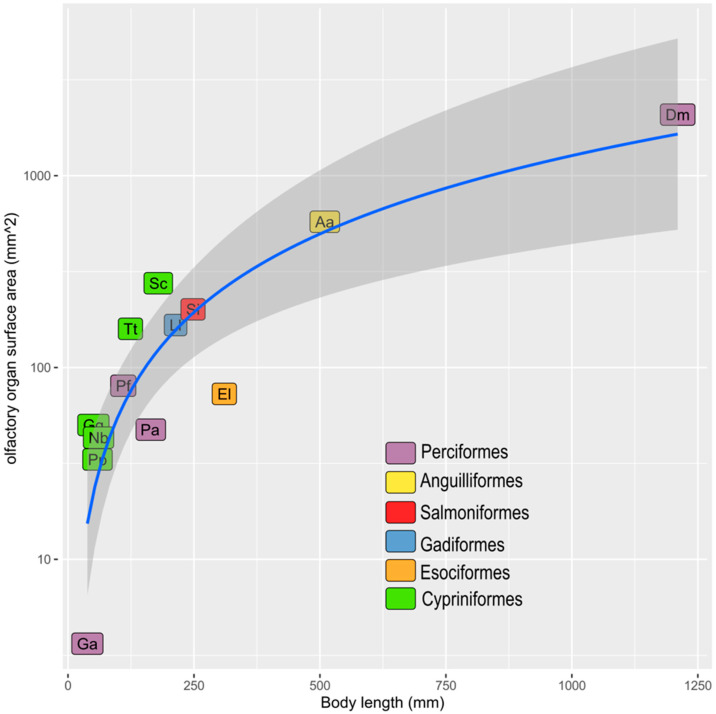
Scatter plot representing data in Table 1. The regression function is y = 0.0524 × 1.3654 (R2 = 0.9756). Aa: *Anguilla anguilla*, Dm: *Dissostichus mawsoni*, El: *Esox lucius*, Ga: *Gasterosteus aculeatus*, Gg: *Gobio gobio*, Ll: *Lota lota*, Nb: *Nemachilus barbatulus*, Pa: *Pleuragramma antarcticum*, Pf: *Perca fluviatilis*, Pp: *Phoxinus phoxinus*, Sc: *Squalius cephalus*, Si: *Salmo irideus*, Tt: *Tinca tinca*.

**Figure 6 animals-12-00663-f006:**
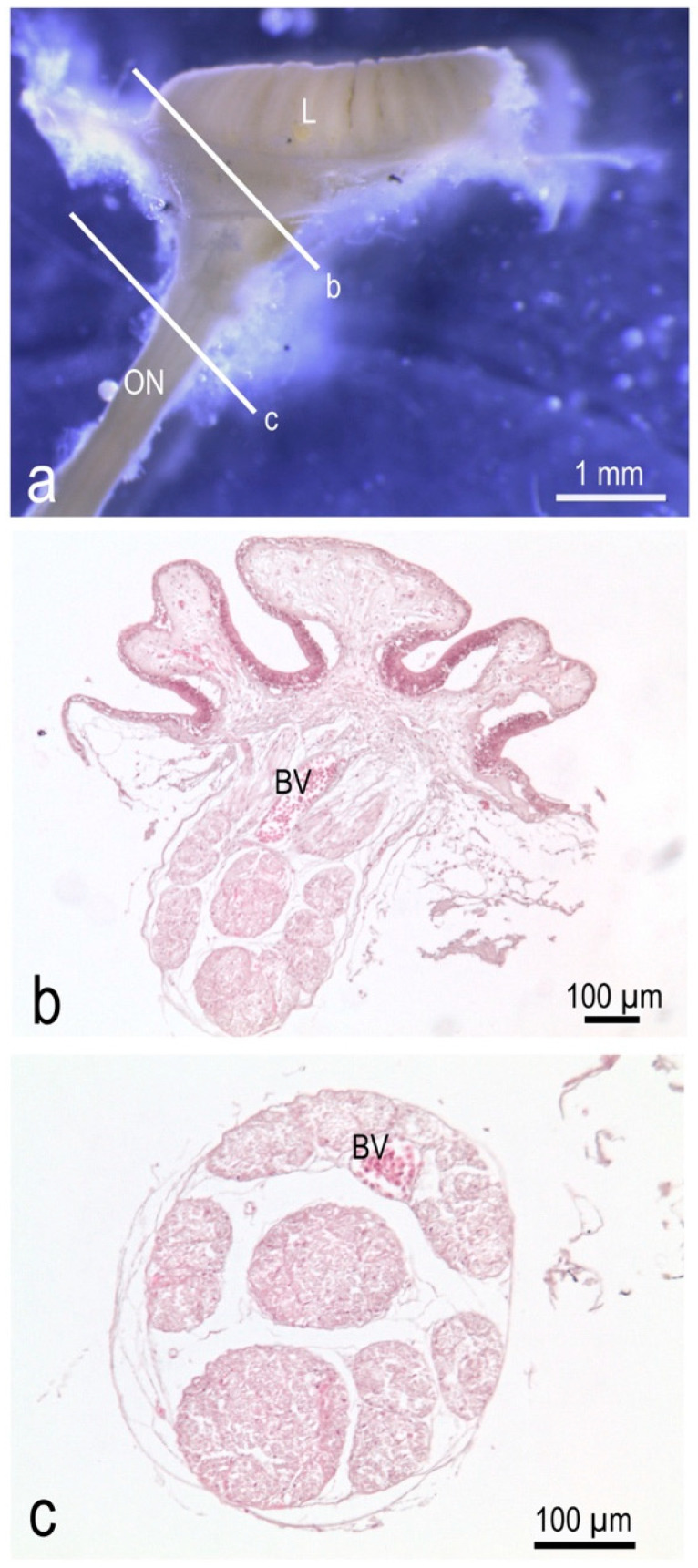
Gross morphology and histology of the olfactory rosette and nerve of *P. antarcticum*. (**a**) Lateral view of an olfactory rosette with the olfactory nerve. The two lines indicate the cutting plane of histological photographs in (**b**,**c**). (**b**) Hematoxylin-Eosin; the fila olfactoria gather to form the olfactory nerve in the connective tissue under the raphe. The large blood vessel that runs along the raphe, follows the fila olfactoria. (**c**) Hematoxylin-Eosin; transverse section of the olfactory nerve. BV = blood vessel; L = lamellae; ON = olfactory nerve.

**Figure 7 animals-12-00663-f007:**
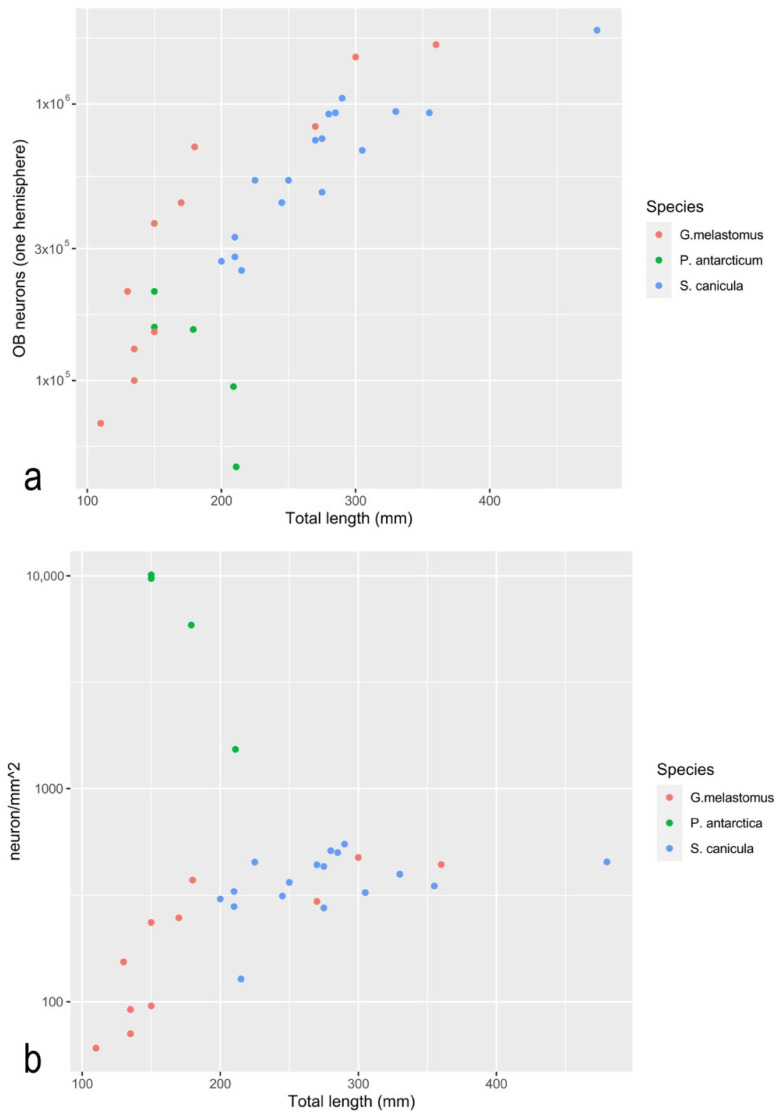
Scatterplots representing part of data in Table 2. Quantitative data from olfactory rosette and bulb of *P. antarcticum* (present work), and *G. melastomus* and *S. canicula* [51]. (**a**) Scatterplot showing the number of neurons in one olfactory bulb. (**b**) Scatterplot showing the number of neurons in one olfactory bulb normalized to the mm^2^ of surface area of the olfactory organ.

**Table 1 animals-12-00663-t001:** Measurements and counts for 13 species of teleost fish from the present study and from the literature. SL = standard length; TL = total length; the body size reported in the table is SL for all the species—except *D. mawsoni,* for which the total length is indicated. Whole ES = epithelial surface area for two olfactory organs—that is, the whole surface of the fish; ES = epithelial surface area of one olfactory organ.

Order	Species	SL or TL (mm)	Whole ES (mm^2^)	ES (mm^2^)	Source
Anguilliformes	*Anguilla anguilla*	509.6	575.16	287.58	Teichmann, 1954 [17]
Cypriniformes	*Phoxinus phoxinus*	58.58	33.15	16.575	Teichmann, 1954 [17]
Cypriniformes	*Gobio gobio*	50.41	50.16	25.08	Teichmann, 1954 [17]
Cypriniformes	*Squalius cephalus*	179.2	275.09	137.545	Teichmann, 1954 [17]
Cypriniformes	*Tinca tinca*	123.93	159.09	79.545	Teichmann, 1954 [17]
Cypriniformes	*Nemachilus barbatulus*	60.64	43.09	21.545	Teichmann, 1954 [17]
Esociformes	*Esox lucius*	310.2	72.82	36.41	Teichmann, 1954 [17]
Gadiformes	*Lota lota*	213.5	166.532	83.266	Teichmann, 1954 [17]
Perciformes	*Perca fluviatilis*	109.62	80.35	40.175	Teichmann, 1954 [17]
Perciformes	*Gasterosteus aculeatus*	38.48	3.64	1.82	Teichmann, 1954 [17]
Perciformes	*Pleuragramma antarcticum*	158.4	47.42	23.71	present paper
Perciformes	*Dissostichus mawsoni*	121	2082	1041	Ferrando et al. 2019 [16]
Salmoniformes	*Salmo irideus*	248.15	201.01	100.505	Teichmann, 1954 [17]

## Data Availability

The data presented in this study are openly available in FigShare at https://doi.org/10.6084/m9.figshare.19114934 and https://doi.org/10.6084/m9.figshare.19126940.v1 (accessed on 3 March 2022).

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
