# Peer review of "The Arrangement of the Peripheral Olfactory System of Pleuragramma antarcticum: A Well-Exploited Small Sensor, an Aided Water Flow, and a Prominent Effort in Primary Signal Elaboration"

_animals, 2022, doi:10.3390/ani12050663_

Round 1

Reviewer 1 Report

This ms reports morphometric data of the peripheral olfactory system (i.e., rosette mucosa and olfactory nerve) in an antarctic fish, Pleuragramma antarcticum (n=6).

This is a comparative anatomy paper that elegantly reflects the difficulties in comparing homologous structures across different species, which have been examined previously with different methods.

The data are carefully collected and documented and may be a useful tool once physiologic investigations on the chemical senses are being done.

Although mainly descriptive, the authors try to speculate on some functional issues, and, as expected, with no physiologic/behavioral data etc at hand, the conclusions (last sentence of simple summary and lines 442f) appear predictably poor. The notion that pollution might harm these species, appears somewhat trivial without indication, which fish could not be affected.

Minor critical points:

  1. The Abstract does not really reflect the research topic, and neither the Title. It would be helpful to shift some of the Simple Summary into the Abstract section, as the content of the former does not appear in general indices such as Pubmed.
  2. Mitral cells are "projection neurons" rather than "large interneurons". Interneurons at the OB level are typically periglomerular cells, which, however, seem not to occur in fish glomeruli.

Reviewer 2 Report

This paper is showing the morphology of the olfactory system of Antarctic silverfish. Very interesting and I enjoyed reading it. It is a rare observation of this species of fish, I believe, and comparison with other fish species is very interesting. I am sure that readers will enjoy reading this paper.

One thing that I wondered was how the authors selected the other fish species to compare with the Antarctic silverfish. Was it based on a practical reason (availability of data) or based on the similarity in ecological habitat or selected from the studies on rather closely relatedness species that were available. Although the authors did add some description on natural habitat as well as the characteristics and life stage changes in the habitat of Antarctic silverfish, I think it is nice to briefly add information of those of the fish species that were selected to compare with Antarctic silverfish, in the Methods and in the Discussion.

Minor comments:

Line 158: Add the centrifuge speed and time length

Line 177: remove “(“

Table 1: It is better to add the information on the statuses of the fish whether they were fresh, fixed, what they were fixed with if they were fixed because fixation affects the size.

Line 251: Figure 4 -> In the figure it says Figure 3, and the caption of Figure 3a says the location is indicated as the third line in Figure 2.

Figure 2a: It will be helpful to add a thin dotted line to show where the olfactory rosette is, where the ON runs, and where the OB is if the OB is included in the photo.

Line 256: what do you mean by “lamellae, at the sides, are free”?

Figure 2c: so this means that there are part of the lamellae that are located anterior to the nostril?

Figure 3: very pretty

Figure 3a: Although it is easy to assume, it will be helpful for readers if arrows in + shape, showing vertical and horizontal direction are added.

Line 270: “is the smallest” -> they are all connected, isn’t it? Any reason it is necessary to say the size is smallest here in the view??

Line 271: being divide -> being divided

Line 294: Table 1 -> Table 2

Line 285~: This is a simple question from a viewpoint of non-fish person, but how precise this measurement can be? They have curved areas that it is necessary to estimate how larger it becomes when the curved surface is spread from the size of the lamellae. Did you measure using ImageJ/Fiji using the images like these in Figure 4? Or did you measure macroscopicly the lamellae using a caliper and estimated the surface size?

Figure 5: Add in the caption what each abbreviation (Ga, Pa, El, and so on) in the figure means so that readers will not need to go to Table 1 to confirm.

Line 325: remove the “,” before (Figure 6c).

Line 349-350: I believe this is remaining of the Instruction to Authors. Remove it.
